# Health disparities and COVID-19: A retrospective study examining individual and community factors causing disproportionate COVID-19 outcomes in Cook County, Illinois

**Larissa H. Unruh[1]◉, Sadhana Dharmapuri[2,3]◉, Yinglin Xia [4]◉, Kenneth Soyemi [2,3]◉ ***

**1** Department of Emergency Medicine, John H. Stroger Jr. Hospital of Cook County Health, Chicago, Illinois, United States of America, **2** Cermak Health Services, Cook County Juvenile Temporary Detention Center, Chicago, Illinois, United States of America, **3** Department of Pediatrics, John H. Stroger Jr. Hospital of Cook County, Chicago, Illinois, United States of America, **4** Department of Medicine, University of Illinois at Chicago, Chicago, Illinois, United States of America

◉ These authors contributed equally to this work.
* ksoyemi@cookcountyhhs.org

**Data Availability Statement:** All data used in this manuscript are publicly available online. Data can be accessed through: SOYEMI, KENNETH (2022),

## Abstract

Early data from the COVID-19 pandemic suggests that the disease has had a disproportionate impact on communities of color with higher infection and mortality rates within those communities. This study used demographic data from the 2018 US census estimates, mortality data from the Cook County Medical Examiner's office, and testing results from the Illinois Department of Public Health to perform bivariate and multivariate regression analyses to explore the role race plays in COVID-19 outcomes at the individual and community levels. We used the ZCTA Social Deprivation Index (SDI), a measure of ZCTA area level deprivation based on seven demographic characteristics to quantify the socio-economic variation in health outcomes and levels of disadvantage across ZCTAs. Principal findings showed that: 1) while Black individuals make up 22% of Cook County's population, they account for 28% of the county's COVID-19 related deaths; 2) the average age of death from COVID-19 is seven years younger for Non-White compared with White decedents; 3) residents of Minority ZCTA areas were 1.02 times as likely to test positive for COVID-19, (Incidence Rate Ratio (IRR) 1.02, [95% CI 0.95, 1.10]); 1.77 times as likely to die (IRR 1.77, [95% CI 1.17, 2.66]); and were 1.15 times as likely to be tested (IRR 1.15, [95% CI 0.99, 1.33]). There are notable differences in COVID-19 related outcomes between racial and ethnic groups at individual and community levels. This study illustrates the health disparities and underlying systemic inequalities experienced by communities of color.

## Introduction

On March 11, 2020, the World Health Organization officially declared COVID-19, the disease caused by the SARS-CoV-2 virus, a pandemic. The virus has infected over 200 million people

"Cook County Medical Examiners Mortality Data", Mendeley Data, V1, doi: 10.17632/grwgy44rwj.1 (https://doi.org/10.17632/grwgy44rwj.1).

**Funding:** The authors received no specific funding for this work.

**Competing interests:** The authors have declared that no competing interests exist.

worldwide and caused approximately 4.5 million deaths. The United States has become the epicenter of this pandemic with an incidence of 43 million cases and approximately 700,000 or 16% of the reported global deaths [1]. The US has until recently experienced rapid, largely unchecked disease spread with subsequent increases in infections and mortality.

Data from the Centers for Disease Control and Prevention (CDC) as well as many subsequent studies and articles suggests that the burden of COVID-19 morbidity and mortality is not uniformly distributed throughout the US population [2–5]. People of color, especially those within Black and Latin X communities, are disproportionately affected by the disease. These disparities have been widely discussed by media outlets, medical journals, health associations, and acknowledged by political leaders [2–5]. While scientific literature supports the idea that societal norms that systematically disadvantage people of color (i.e., structural racism) play a major role in disease outcome, that knowledge has yet to become a widely acknowledged fact in society. Consequently, few recent studies address social determinants of health (SDH), such as differences in access to housing, transportation, and food as well as representation within correctional facilities as contributing factors for disparate disease outcomes between communities of color and White communities [2–7]. Half of deaths in the US are related to personal behaviors that are propelled by social factors, including income, education, and employment. Socioeconomic gradients are present among all health indicators [8]. There is a causal link between social factors, health indicators, and measures of individuals' socioeconomic resources, such as income, educational attainment, and social position [9]. This association simulates a stepwise gradient pattern, with health improving incrementally as social position rises. This relationship is dose-dependent and is predominant in Non-Hispanic Black and Non-Hispanic White groups compared with Latin X populations. Social determinants of health contribute to disparities because differences that exist in life expectancy between racial and ethnic groups are socially and culturally established, not biologically derived [10]. Long term exposure to stress induced by social and environmental factors cause biological "wear-and-tear" also known as allostatic load [11]. Epigenetic processes that regulate whether genes are expressed or suppressed is one biological mechanism involved in the linkage of social factors to health outcomes. Primate studies indicate that social status can affect the regulation of genes that control physiologic functions (e.g., immune functioning). Educational attainment, occupational class (e.g., manual vs. non-manual work), work schedules, stress, and intimate partner violence are linked with changes in DNA-protein complexes capping the ends of chromosomes (telomeres) [12].

Despite many rapid advances in the scientific understanding of COVID-19, much remains unknown about why certain people experience worse outcomes than others. However, as is true with many diseases, COVID-19 outcomes are strongly associated with race [13–17]. A systematic review by Mackey et al. [18] explored outcome differences between races and confirmed that Black and Latinx individuals suffered higher rates of infection, hospitalization, and mortality than White individuals. The Mackey study attributes these differences to differences in health care access and exposure risk. In fact, based on their thorough literature review, Blacks have between 1.5- and 3.5-times risks of infection, hospitalization, and dying with 15% excess mortality when compared with Whites. Similarly, the same study showed that Latin X have between 1.3 and 7.7 times risks of infection, hospitalization, and dying with 21% excess mortality also when compared with Whites [18].

In "Racial Health Disparities and Covid-19 –Caution and Context," published in *The New England Journal of Medicine*, Chowkwanyun and Reed discussed the need for careful data analysis before making blanket statements about racial differences in COVID-19 morbidity and mortality [19, 20]. In keeping with Chowkwanyun and Reed's recommendations, our study describes the demographic distribution of mortality and explores whether there is an

association between race, zip code tabulation areas (ZCTAs), and COVID-19 outcomes in Cook County, Illinois. Additionally, we explore five potential social determinants of health which may contribute to the disparities in COVID-19 mortality data.

## Materials and methods

### IRB approval

This investigation was conducted as part of public health practice and was classified as exempt by the Cook County Health Institutional Review Board. We performed retrospective cohort analysis to explore COVID-19 outcomes between March 2020, and September 2021. We used three data sources to create two master spreadsheets to analyze both individual and ZCTA-level aspects of COVID-19 morbidity and mortality.

### Individual level analysis

The Cook County Medical Examiner's website provided COVID-19 mortality data (Accessed October 1, 2021 [20]. Data elements included decedent demographics (age sex race/ethnicity zip code), primary and secondary causes of death, and date and location of death. Descriptive statistics for continuous variables were presented as means ± standard deviations (SDs). Categorical variables were presented as frequencies and proportions. All statistical tests were two-sided, and p-values <0.05 were considered statistically significant. For baseline characteristics, after determining whether continuous variables were normally distributed, parametric methods (independent *t*-test and ANOVA) and non-parametric (*Wilcoxon rank sum test* and *Kruskal-Wallis test)* were used to determine whether differences in groups were significant. Comorbidities were aggregated for supplemental analysis. To determine the impact of the combination of comorbidities with COVID-19 between Non-White (Black, Latin X, Asian, and other race decedents) vs White; we used the *chi*-square test of independence to determine differences between two groups. We also reported the Relative Risk (RR) and risk differences and their corresponding 95% confidence intervals.

### ZCTA analysis

The Illinois Department of Public Health (IDPH) provided zip code level numbers for COVID-19 tests completed and positive cases (Accessed October 1, 2021). Decedents' zip codes were cross walked with ZCTAs, which are spatial representations of US Postal Service zip code service areas, and in Chicago represent the same spatial areas. The 2018 US Census Bureau's 5-year estimate for Cook County, Illinois, provided ZCTA level demographic data. We used the ZCTA Social Deprivation Index (SDI), a measure of ZCTA area level deprivation based on seven demographic characteristics collected in the American Community Survey (https://www.census.gov/programs-surveys/acs/). The SDI is used to quantify the socio-economic variation in health outcomes and levels of disadvantage across ZCTAs. The seven demographic characteristics are: percent living in poverty, percent with less than 12 years of education, percent single parent households, percent living in rented housing units, percent living in overcrowded housing units, percent of households without a car, and percent non-employed adults underage 65 [21]. We added average household size, average per capita income, percent of households that spoke English and Spanish, percent of households with bachelor's degrees as additional covariates. Mortality data after aggregation by ZCTA were merged with ZCTA level SDI and additional ZCTA-level census data described above to create the ZCTA-level data set. Minority population was defined as any race or ethnic group other than Non-Hispanic White (White). Minority predominant or Non-White

ZCTAs (Minority Zip) were defined as those in which the minority population was above 50% of the total population.

## Statistical analyses

To explore the relationship between COVID-19 mortality and contributing ZCTA factors, a count modeling strategy including Poisson, negative binomial (NB), zero-inflated Poisson (ZIP), and zero-inflated negative binomial (ZINB) was used to model the main outcomes of interest (e.g., number of deaths, number of positive tests, and number of residents tested). We compared the predictive performance of Poisson, NB, ZIP, and ZINB models using likelihood ratio, Akaike's information criterion (AIC), Bayesian information criterion (BIC), standardized Pearson residual, and the standardized deviance residuals. In addition, differences between the predicted and actual counts were compared based on the mean squared error (MSE) performance measure.

For ZCTA analysis, we created one model for each outcome variable (mortality, number of positive tests and total number tested per ZCTA). For the mortality (deaths) per ZCTA, we used four models with sequential addition of economic and housing population covariates not included in the SDI. To understand the relationships between the variables we added the interaction term of Non-White (Minority Zip) ZCTA and if greater than 20% of households lived in poverty. Our outputs were the incident rate ratios (IRR) and the predicted probabilities of number of deaths, number of tests, and number of positive results adjusted for the social deprivation score and the added covariates, using margins (Stata command). Margins are statistics calculated from predictions of a previously fit model at fixed values of some covariates and averaging or otherwise integrating the remaining covariates. The "margins" command estimates margins of responses for specified values of covariates and presents the results as a table. Each predicted margin is computed by fixing the SDI score at a given value (20 to 100 in increments of 10) and leaving all other covariates as they were in the model. The predicted values of the outcome variable are averaged and plotted using the "marginsplot" command.

The figures were created using Microsoft Excel, and statistical analysis was performed using SAS software (version 9.4; SAS Institute) and Stata (version 17; Stata Corp LLC).

## Results

Cohort sample size included 10,813 COVID-19 related mortalities. Of the 10,813 deaths, 4,527 (42%) were White, 3,224 (29%) Black, 2,335 (22%) Latin X, 435 (4%) Asian, and the rest were classified as Other. Males accounted for 6,142 (57%) and 8,018 (75%) were persons over age 65. Persons 85 years of age and older had the highest mortality, 2,812 (26%). Overall, mean (SD) age at death was 73 (15) years. Mean (SD) age at death for males compared with females was, 71.2 (15) vs 76.2 (15); $p < 0.01$. The Mean (SD) age at death by race and ethnicity was White 78 (13), Black 71 (15), Latinx 67 (15), Asian 77 (14), and other 75 (12); p <0.01. Hypertension was the most frequently recorded comorbidity 6,480 (60%) among decedents (Table 1).

Mean age at death was lower for Non-Whites compared with Whites 70 (15) vs 78 (13); p < 0.01. Non-White decedents had higher prevalence of diabetes, obesity, and hypertension compared with White decedents. The number of comorbidities recorded was 0–9, with 2 being the most reported 3,224 (30%). For both Non-Whites and Whites, a higher proportion of males died compared with females 3,632 of 6,227 (58%) vs. 2,485 of 4,527 (55%); risk difference of 0.03 [(95% CI: 0.01, 0.05); P = <0.01)]. When COVID-19 was associated with a comorbidity, risk differences of Non-Whites compared with Whites were 0.13 [(95% CI: 0.11, 0.15); P = <0.01)] for diabetes; 0.01 [(95% CI: -0.07, 0.03); P = <0.20)] for hypertension; and 0.04

**Table 1. Population and COVID-19 mortality, Cook County, March 15-May 31, 2021.**

| Cook County general information | | | Total |
|---|---|---|---|
| | Total population | 5,569,118 | |
| | Total households | 2,084,482 | |
| | Average household size | 2.67 | |
| | Land area (Miles) | 1,118.31 | |
| | Population density (per mile$^2$) | 4,979.94 | |
| | Average per capita income | $37,538 | |
| | Median age (Years) | 37 | |
| **Cook County population by sex** | | | **5,569,118** |
| | Female | 2,864,819 (51%) | |
| | Male | 2,704,299 (49%) | |
| **Cook County population by race** | | | **5,569,118** |
| | Black | 1,227,577 (22%) | |
| | Latin X | 1,354,630 (24%) | |
| | White | 2,469,254 (44%) | |
| | Asian | 379,016 (6%) | |
| | Other | 138,641 (4%) | |
| **Mortality by age** | | | **10,813** |
| | Age range | 0–109 | |
| | Mean (SD) age | 74 (15) | |
| | Median (IQR)age | 75 (65–75) | |
| **Mortality by sex** | | | **10,750** |
| | Male | 6,117 (57%) | |
| | Female | 4,633 (43%) | |
| **Mortality by race/ethnicity** | | | **10,777** |
| | White | 4,527 (42%) | |
| | Black | 3,224 (29%) | |
| | Latin X | 2,335 (22%) | |
| | Asian | 435 (4%) | |
| | Other | 256 (2%) | |
| **Mortality by Age Group** | | | **10,813** |
| | Less than 15 years | 7 (0%) | |
| | 15–24 years | 30 (0%) | |
| | 25–34 years | 132 (1%) | |
| | 35–44 years | 304 (3%) | |
| | 45–54 years | 722 (7%) | |
| | 55–64 years | 1,599 (15%) | |
| | 65–74 years | 2,574 (24%) | |
| | 75–84 years | 2,632 (24%) | |
| | 85 years and above | 2,812 (26%) | |
| **Number of comorbidities reported per decedent** | | | **10,813** |
| | 0 | 1,664 (15%) | |
| | 1 | 2,826 (26%) | |
| | 2 | 3,224 (30%) | |
| | 3 | 1,775 (16%) | |
| | 4 | 892 (8%) | |
| | 5 | 304 (3%) | |
| | 6 | 90 (0.8%) | |

(*Continued*)

**Table 1.** (Continued)

| Cook County general information | | | Total |
|---|---|---|---|
| | 7 | 34 (0.3%) | |
| | 8 | 3 (0.03%) | |
| | 9 | 1(0.1%) | |
| **Comorbidities** | | | **10,813** |
| | Hypertension | 6,480 (60%) | |
| | Diabetes | 4,467 (41%) | |

[(95% CI: 0.03, 0.05); P = <0.01]] for obesity. When hypertension, obesity, and diabetes were combined, the risk difference was 0.02 [(95% CI: 0.01, 0.03); P = <0.01))]. Direct comparison of COVID-19 associated morbidities for Non-White compared with White decedents with hypertension showed no difference [Relative Risk (RR) 1.01, 95% CI (0.98, 1.05)]. However, risk rates were higher for Non-Whites with diabetes [RR = 1.39, 95% CI (1.32. 1.46)], obesity [RR = 1.60, 95% CI (1.42. 1.80)], and the additive combination of hypertension, obesity, and diabetes [RR = 1.74, 95% CI (1.41. 2.14)] (Table 2).

One hundred and sixty-four Cook County ZCTAs recorded deaths, tests, and positive results. Of the 164 ZCTAs, 76 (46%) were minority predominant ZCTAs (Minority Zip). Overall, mean (SD) number of deaths, number tested, number with positive test results, and

**Table 2. Comparison of mortality and co-morbidities White Vs Non-White, Cook County, Illinois.**

| Factor | Level | Non-White N = 6227 N (%) | White N = 4527 N (%) | ALL N = 10754 N (%) | Risk Difference (95% CI) | Relative Risk (95% CI) | *P*-value* |
|---|---|---|---|---|---|---|---|
| Sex | Male | 3632 (58) | 2485 (55) | 6117 (57) | 0.03 (0.02,0.05) | 1.06 (1.02,1.09) | < 0.01 |
| | Female | 2593 (42) | 2040 (45) | 4633 (43) | | | |
| Hypertension | Yes | 3761 (60) | 2683 (59) | 6444 (60) | 0.01 (-0.01,0.03) | 1.01 (0.98,1.05) | 0.20 |
| | No | 2466 (40) | 1844 (41) | 4310 (40) | | | |
| Diabetes Mellitus | Yes | 2921 (47) | 1522 (34) | 4443 (41) | 0.13 (0.11,0.15) | 1.39 (1.32,1.46) | < 0.01 |
| | No | 3306 (53) | 3005 (56) | 6311 (59) | | | |
| Chronic Obstructive Pulmonary Disease | Yes | 588 (9) | 736 (16) | 1324 (12) | -0.06 (-0.08, -0.05) | 0.58 (0.52,0.64) | < 0.01 |
| | No | 5639 (91) | 3791 (84) | 9430 (88) | | | |
| Hypertensive Coronary Vascular Disease | Yes | 247 (4) | 152 (3) | 399 (4) | 0.01 (-0.00,0.13) | 1.18 (0.96,1.44) | 0.09 |
| | No | 5980 (96) | 4375 (97) | 10355 (96) | | | |
| Coronary Artery Disease | Yes | 712 (11) | 783 (17) | 1495 (14) | -0.05 (-0.07, -0.04) | 0.66 (0.60,0.73) | < 0.01 |
| | No | 5515 (89) | 3744 (83) | 9259 (86) | | | |
| Dementia | Yes | 395 (6) | 490 (11) | 885 (8) | -0.04 (-0.05, -0.03) | 0.58 (0.52,0.67) | < 0.01 |
| | No | 5832 (94) | 4037 (89) | 9869 (92) | | | |
| Atrial Fibrillation | Yes | 406 (7) | 676 (15) | 1082 (10) | -0.08 (-0.09, -0.07) | 0.43 (0.38,0.49) | < 0.01 |
| | No | 5821 (93) | 3851 (85) | 9672 (90) | | | |
| Congestive Heart Failure | Yes | 527 (8) | 572 (13) | 1099 (10) | -0.04 (-0.05, -0.02) | 0.67 (0.61,0.75) | < 0.01 |
| | No | 5700 (92) | 3955 (87) | 9655 (90) | | | |
| Chronic Kidney Disease | Yes | 641 (10) | 581 (13) | 1222 (11) | -0.02 (-0.03, -0.01) | 0.80 (0.72,0.89) | < 0.01 |
| | No | 5586 (90) | 3946 (87) | 9532 (89) | | | |
| Obesity | Yes | 764 (12) | 348 (8) | 1112 (10) | 0.04 (0.03,0.05) | 1.60 (1.42,1.80) | < 0.01 |
| | No | 5463 (88) | 4179 (92) | 9642 (90) | | | |
| Combination of Hypertension Diabetes and Obesity | Yes | 290 (5) | 121 (3) | 411 (4) | 0.02 (0.01,0.03) | 1.74 (1.41,2.14) | < 0.01 |
| | No | 5937 (95) | 4406 (97) | 10343 (96) | | | |

total population per ZCTA was 65 (59), 71,692 (46,473), 4,008 (2,979), and 34,043 (21,633). When Minority Zip was compared with Non-Minority Zip, mean (SD) for the number of deaths, number tested, number with positive test results, and total population were 85 (66), 79,976 (50,599), 4,919 (3,722), and 39,133 (25,816) vs 48(44), 65,401 (41,940), 3,222(1,826), and 29,647 (16,127). The mean (SD) social deprivation index score for Minority Zip was higher compared with Non -Minority Zip (74 (23) vs 32 (22); $P$ value <0.01)). NB regression showed that residents of Non-White ZCTAs were 1.02 times as likely to test positive and 1.15 times as likely to get a test when compared with residents of predominantly White ZCTAs [Incident Rate Ratio (IRR) 1.02, 95% CI (0.95, 1.10) and IRR 1.15, (0.99, 1.33)]. These rates were adjusted for population density, average household size, SDI score, per capita income, households living below the federal poverty line. When the number of deaths, positive tests, and number tested were modelled with no covariates, residents of Non-White ZCTAs were 1.67 times [IRR 1.62, 95% CI (1.23, 2.12)] more likely to die, 1.02 times as likely to be positive [IRR 1.02, 95% CI (0.81,1.02)] and 1.15 times as likely to be tested [IRR 1.15, 95% CI (0.99, 1.33)]. In the final model after adjusting for factors described above, residents of Non-White ZCTAs were 1.77 times as likely to die from COVID-19 [IRR 1.77, 95% CI (1.17, 2.66)]. Residents in households with a high percentage of Spanish speakers had increased risk of testing positive [IRR 1.69, 95% CI (1.03, 2.75)]. Living in poverty alone was not associated with an increased number of deaths, but the interaction of living in poverty and in a Non-White ZCTA increased the risk 2 to 3 times [IRR 2.99, 95% CI (0.71, 12.57)]. (Table 3).

For every one-point increase in SDI score the number of deaths, number of tests, and number that tested positive all increased one unit. Using the "margins" command and fixing the SDI score at 20 to 100 in increments of 10, the predicted number of deaths increased as the SDI score increase for both Non-White ZCTA and White ZCTA, but Non-White ZCTAs had a higher slope (Fig 1).

**Table 3. Negative binomial regression model outputs.**

| | Model 1 | Model 2 | Model 3 | Model 4 | Model 5 | Model 6 | Model 7 |
|---|---|---|---|---|---|---|---|
| | Deaths IRR 95%CI | Deaths IRR 95%CI | Deaths IRR 95%CI | Deaths IRR 95%CI | Deaths IRR 95%CI | Tested IRR 95%CI | Positive IRR 95%CI |
| Non-White Predominant ZCTA | 1.62*** [1.23,2.12] | 1.07 [0.75,1.54] | 1.25 [0.86,1.84] | 1.77** [1.17,2.66] | 1.77** [1.17,2.66] | 1.15 [0.99,1.33] | 1.02 [.0.95,1.10] |
| Social Deprivation Score | | 1.01** [1.00,1.02] | 1.01** [1.00,1.02] | 1.01* [1.00,1.02] | 1.01 [1.00,1.01] | 1.00 [1.00,1.00] | 1.00 [1.00,1.00] |
| >20% below the FPL | | 0.47 [0.10,2.13] | 0.34 [0.08,1.42] | 0.38 [0.10,1.48] | 0.41 [0.10,1.60] | 2.01*** [1.36,2.96] | 1.13 [0.83,1.55] |
| Non-White Predominant ZCTA # > 20% below the FPL | | 2.41 [0.50,11.66] | 2.98 [0.67,13.18] | 3.16 [0.76,13.08] | 2.99 [0.71,12.57] | 0.65* [0.43,0.99] | 0.81 [0.58,1.13] |
| Average Household Size | | | 0.59* [0.38,0.92] | 0.60* [0.36,0.99] | 0.54* [0.31,0.96] | 0.74*** [0.64,0.87] | 1.09 [0.96,1.24] |
| Percent Speak English | | | | 0.06*** [0.02,0.18] | 0.06*** [0.02,0.18] | 0.73 [0.48,1.11] | 1.02 [0.75,1.38] |
| Percent Speak Spanish | | | | 0.05*** [0.01,0.25] | 0.06*** [0.01,0.29] | 0.71 [0.38,1.34] | 1.69* [1.03,2.75] |
| Income (Per Capita) | | | | | 1.00 [1.00,1.00] | | |
| lnalpha | 0.35*** | | 0.35*** | | 0.35*** | | 0.35*** |
| AIC | 1564 | 1563 | 1563 | 1557 | 1549 | 3608 | 2496 |
| BIC | 1573 | 1581 | 1585 | 1585 | 1580 | 3636 | 2524 |
| Observations | 164 | 164 | 164 | 164 | 164 | 164 | 164 |

Exponentiated coefficients; 95% confidence intervals in brackets

* $p < 0.05$,

** $p < 0.01$,

*** $p < 0.001$

IRR: Incident rate ratio, HH: Household, AIC: Akaike information criterion, BIC: Bayesian information criterion; FPL: Federal Poverty Lin

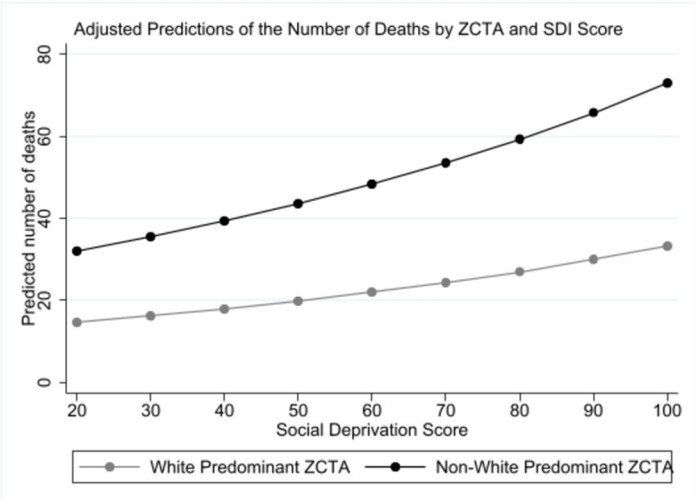

**Fig 1. Predicted number of deaths compared with SDI score.**

## Discussion

As of September 2021, Cook County, Illinois, had identified more than 500,000 positive COVID-19 cases and over 10,000 COVID-19 related deaths [22]. Completion of data analysis reveal that as with national COVID-19 trends, data from Cook County suggest that communities of color had a disproportionate number of cases and deaths. In April 2020, Blacks accounted for 36% of the county's COVID-19 related deaths despite making up 22% of Cook County's population. A year later Blacks accounted for 28% of the deaths confirming persistent disproportionate mortality among Blacks (Fig 2). In the absence of complete state-level data regarding incidence of COVID-19 stratified by race and ethnicity, ZCTA level data analysis displays the burden of disease and impacts of COVID-19 diagnoses and deaths among residents living in Non-White predominant ZCTAs. Modelling with single and multiple covariates, our analyses confirmed persistent higher incident rates of COVID-19 deaths among Non-Whites. As outlined in other studies, our data showed that minorities are dying of COVID-19 at significantly younger ages than Whites (median Non-White age = 72, median White age = 79; p < 0.01).

The SDI uses data on neighborhood social determinants of health, to model health outcomes and the use of health services to study the stability of the model across different geographical areas, such as counties, census tract, and ZCTAs [22]. Comparison of mean SDI score for Non-White predominant ZCTAs compared with White predominant ZCTAs showed significant differences. When the SDI score was combined with other co-variates, the rate of deaths, positive tests, and number tested decreased. This finding confirmed the presence of other factors on COVID-19 outcomes and that the effect of social deprivation factors on COVID-19 outcomes might be minimal with other intrinsic factors causing the disproportionate outcomes. A similar study that used area deprivation index (ADI) showed that positive relationship between area deprivation and COVID-19 prevalence, and strength were higher for rural than for urban counties. Family income, property value, and educational attainment were among the ADI component measures most correlated with prevalence, but this too differed between county type [23].

Other studies that used a deprivation index showed that odds of COVID-19 infection among residents living in areas of very high deprivation were three times higher than those of

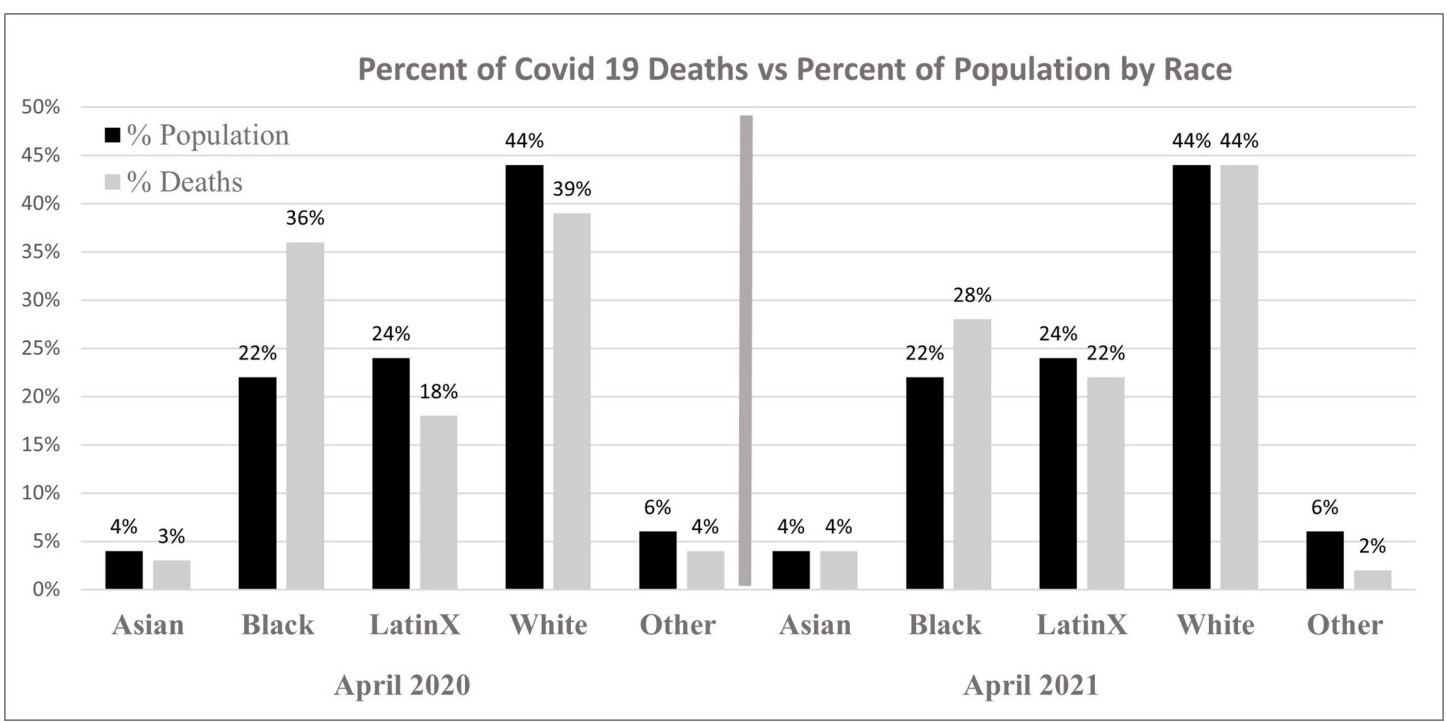

**Fig 2. Percent of COVID-19 deaths vs percent of population by race.**

residents of areas of very low deprivation. Odds of hospitalization among residents of very high-deprivation areas were 1.6 times those among residents of very low-deprivation areas. Likelihood of testing varied less with deprivation than did incidence of hospitalization [5, 24]. Emeruwa et al., showed that risk factors for COVID-19 might cluster within geographic areas. For example, persons living in Non-White predominant ZCTAs with a high deprivation index might be both more likely to work in settings where they could become infected and to live in higher-density settings where household members could become secondarily infected [25]. Therefore, Emeruwa and associates concluded that most of the area-level analysis using some form of deprivation index show a positive correlation with positivity, testing, hospitalization, and mortality.

The association between high social risk and COVID-19 infection should form the basis for the development of preventive strategies to control modifiable factors that impact COVID-19 infection. A previous study by Del Brutto et al. showed that high social risk, as measured by the SDH score, may increase the risk of COVID-19 infection. For every unit increase of the total SDH score, the odds of COVID-19 seropositivity increased 15%, but the study did not model mortality. In contrast, our study modelled the SDI score versus testing, positivity and mortality, and showed that each increased for every point increase in SDI score. Multivariate models in previous studies also demonstrated that the component of SDH most strongly associated with COVID-19 seropositivity was housing, which suggests that lack of basic home facilities may increase the risk of COVID-19 infection [26].

Of the many reasons for COVID-19 disparities between races, we explored five: 1) lower socioeconomic status (SES), 2) increased comorbidities, 3) multigenerational living conditions, 4) high population density in predominantly Black and Latinx ZCTAs, and 5) minority overrepresentation in correctional facilities [27, 28].

## Socioeconomic status

Previous research regarding the effects of poverty is conclusive, but not everyone accepts the effects of income and education on health across the socioeconomic classes. Some have argued that income-health or education-health relationships reflect inverse relationships (i.e., increased sickness leads to decreased income or lower educational achievement) [29]. Khalatbari-Soltani et al. suggest that socioeconomic position plays a crucial role in COVID-19 outcomes and that these factors may be related to the differences in the COVID-19 burden observed between Non-White and White decedents [30]. While it may be true that certain factors, such as decreased access to nutritious food, less sanitary living conditions, or limited access to health care, affect all people with fewer economic resources, our analysis confirmed that poverty by itself was not a significant factor for COVID-19 related death indicating that there are likely other reasons for higher mortality among Non-White populations. Of the factors explored, living in a predominantly Non-White ZCTA was the only statistically significant variable related to death from COVID-19. When poverty was combined with living in a Non-White predominant ZCTA as an interaction term, the risk of death increased 2 to 3 times.

## Increased co-morbidities

Other research has shown that people with certain co-morbidities (e.g., hypertension, diabetes, obesity) have suffered poorer COVID-19 health outcomes than people without underlying conditions. As we compared comorbidities between ethnic/racial groups, we found that minority COVID-19 decedents had higher likelihood of having comorbidities at the time of death (Table 2).

## Multigenerational living

In a study from Italy published in *Eurosurveillance*, Sjödin et al. explored factors necessary to mitigate the COVID-19 pandemic. The authors found that small household size was necessary for limiting the spread of the disease [31]. In Chicago, the areas with higher household density are primarily located in low density Black and Latinx ZCTAs. Our analysis confirms that larger household size is a contributing factor to COVID-19 mortality. Among all combined cases, for each one person increase in household size there is a 66% increase in mortality.

## Densely populated urban areas

Given that the virus spreads primarily through respiratory/droplet transmission, close contact with an infected individual is required for the virus to spread. It seems logical to assume that more densely populated ZCTAs have higher rates of infection and worse outcomes [32, 33]. In Cook County, however, the more densely populated areas are the city center areas that are predominantly inhabited by wealthy, White individuals. Better outcomes in the highly populated areas of Chicago may well be associated with increased access to healthcare because both proximity to hospitals, and financial means to seek medical assistance. At least in Chicago, infection and mortality rates seem to be more related to the density within households than overall population density.

## Overrepresentation in correctional facilities

Earlier in the pandemic, the Cook County Department of Corrections, one of the largest jail systems in the country, gained media attention about a large COVID-19 outbreak among both the facility's employees and inmates [34]. As with many prisons, it has an over-representation

of minority inmates. As of July 1, 2020, the Cook County Sheriff's Department estimated that around 75% of Cook County inmates were Black.

The demographics of the ZCTA that houses the Cook County Jail (60608) show that 55.4% of its inhabitants are male. This over-representation of the male population may be due to the presence of the prison system. The average number of people per household is 3.09, and the median age is 32.4 years. The population of the ZCTA is largely Latin X (50.7%) and Black (17.24%) with an average income of $21,447 (well below Illinois average household income). The COVID-19 data for the 60608 ZCTA shows that it ranks 19[th] out of 164 for the most COVID-19 deaths and 12[th] for the most positive cases in Cook County. Incidence analysis showed that of the 140 deaths in the 60608 ZCTA, 86 (61%) were Latin X, and 20 (14%) were Black. Therefore, although there is an over-representation of Black individuals within the jail, and the ZCTA 60608 had one of the highest incidents of COVID-19 positives, mortality data does not show that this ZCTA was an outlier area for COVID-19 mortality.

## Limitations

This study had several limitations. First, while data was available about individual mortality cases and broad ZCTA demographics, we lacked information about the demographics of those who did not die or who never tested positive for COVID-19, and we were unable to perform a case-control analysis. Second, we used ZCTA demographic data as a proxy for an individual's circumstances. Because that data was essentially an average, not everyone in that ZCTA was accurately represented. Third, because the study focused on Cook County, it may be generalizable only to large urban counties with high levels of racial and ethnic diversity. Fourth, census data is never able to capture the entirety of a population and may miss undocumented individuals. Fifth, because all death data was obtained from the Cook County Medical Examiner's office, the completeness of that data depended on the Examiner's reporting. Sixth, since the analysis relied on ZCTA level data, and ZCTAs do not remain constant over time, it was subject to the inherent issues associated with small area analyses. Seventh, we do not currently have data to indicate whether some ZCTAs with high mortalities had an overrepresentation of congregate care facilities such as long-term care or rehabilitation facilities, which might have skewed the data. Eighth, SDI is a composite measure of deprivation; we do not know which factor or combination of factors in the index are responsible for the association with COVID-19 outcomes. ZCTA-level population and economic data are drawn from estimates before the pandemic, and some measures such as unemployment might have changed more recently.

## Conclusion

While we agree that caution is important when making statements about race and disease outcomes, our data, and the data of other researchers, indicate that the mortality disparities seen between White and Non-White communities are real and that they are likely impacted by underlying structural racism leading to unequal social determinants of health, such as the unequal access to education, resources, and healthcare faced by communities of color. It is time not only to address the disparities that exist between racial/ethnic groups as they relate to COVID-19 but also to address the underlying structural issues that permit these disparities.

Our study suggests that race, more so than SES, is a significant factor for COVID-19 related mortality in Cook County. Other factors being equal, overall, Black individuals are significantly more likely to die from COVID-19, and Latin X individuals are dying at significantly younger ages than Whites. The question of why people of color suffer disproportionate COVID-19 mortality is more complicated than simply identifying disproportionality. These data indicate that many of the deaths were not inevitable but were the byproducts of ingrained

structural inequality resulting in diminished opportunities. We hope that the lessons from this study can help illuminate the persistent inequalities in our country so that, as a society, we can better address these issues. As we confront the subsequent waves of this disease and work to vaccinate as many people, as quickly as possible, we must acknowledge who within our society suffers the most and find ways to focus our efforts to keep those populations as safe and healthy as possible.

## Acknowledgments

Special thanks to Mary and Gail Unruh, Jeff Nogaj, and Dr. Mark Mycyk for their unwavering help and support.

## Author Contributions

**Conceptualization:** Larissa H. Unruh, Sadhana Dharmapuri, Kenneth Soyemi.

**Data curation:** Larissa H. Unruh, Kenneth Soyemi.

**Formal analysis:** Larissa H. Unruh, Sadhana Dharmapuri, Yinglin Xia.

**Investigation:** Kenneth Soyemi.

**Methodology:** Kenneth Soyemi.

**Writing – original draft:** Larissa H. Unruh, Sadhana Dharmapuri, Yinglin Xia, Kenneth Soyemi.

**Writing – review & editing:** Larissa H. Unruh, Sadhana Dharmapuri, Yinglin Xia, Kenneth Soyemi.

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
