## [Decision Letter · Decision Letter 0]

19 Nov 2020

PONE-D-20-25294

Health Disparities and COVID-19: A Retrospective Study Examining Individual and Community Factors Causing Disproportionate COVID-19 Outcomes in Cook County, Illinois, March 16-May 31, 2020

PLOS ONE

Dear Dr. Soyemi,

Thank you for submitting your manuscript to PLOS ONE. After careful consideration, we feel that it has merit but does not fully meet PLOS ONE’s publication criteria as it currently stands. Therefore, we invite you to submit a revised version of the manuscript that addresses the points raised during the review process.

There clearly is a need to provide better grounding for your research, rearrange some of the sections and attend to the interpretation issues pointed out by the reviewers. Very importantly, all issues regarding the statistical analysis and the inconsistencies pointed out need to be addressed in a revision. It appears that the alignment between methods and actual data and results is not optimal and needs your attention.

We look forward to receiving your revised manuscript.

Kind regards,

Hajo Zeeb

Academic Editor

PLOS ONE

Journal Requirements:

2. Please include the date(s) on which you accessed the databases or records to obtain the data used in your study.

3. We note that Table 1 appears to have a typographical error were one p-value is reported as "<.00" instead of "<.001". Please revise this.

4. PLOS ONE publication criteria and journal policy require authors to make all data underlying the findings described in their manuscript fully available without restriction, with rare exception (https://journals.plos.org/plosone/s/data-availability#loc-acceptable-data-access-restrictions). Please clarify whether all data used in your study is publicly available and how they can be accessed. We encourage authors to share de-identified or anonymized data if possible. For any third-party data that you cannot legally distribute, please include in the Data Availability Statement a description of the data set and all necessary third-party contact information others would need to apply to gain access to the data.

5. In your ethics statement in the Methods section and in the online submission form, please provide additional information about the data used in your retrospective study. Specifically, please ensure that you have discussed whether all data were fully anonymized before you accessed them. If not, please state whether your IRB waived the requirement for informed consent.

7. Please include a caption for figure 1.

Reviewers' comments:

Reviewer's Responses to Questions

**Comments to the Author**

1. Is the manuscript technically sound, and do the data support the conclusions?

Reviewer #1: Yes

Reviewer #2: Partly

2. Has the statistical analysis been performed appropriately and rigorously? 

Reviewer #1: Yes

Reviewer #2: Yes

3. Have the authors made all data underlying the findings in their manuscript fully available?

Reviewer #1: Yes

Reviewer #2: No

4. Is the manuscript presented in an intelligible fashion and written in standard English?

Reviewer #1: Yes

Reviewer #2: Yes

5. Review Comments to the Author

Reviewer #1: Summary:

The authors investigate a very relevant question, the association between minority status and different Covid-19 related outcomes. The authors did a great work in integrating different data sources from the second largest county in the US. However, I see certain analyses that can be improved and do not agree in all their interpretations of the results. Specific comments and concerns are outlined below.

Comments:

1. Page 3 (Line 64). The authors say that they explore a causal relationship. I think a bit more cautious wording should be used with their study design.

2. Page 4. I would find it helpful to make it even clearer (maybe subheadings) in which analyses the unit of investigation are persons and in which ZCTA´s.

3. Page 4 (Line 84). Why was the cut-off of 50% used for the ZCTA´s?

4. Page 4 (Line 86). Moralities -> Mortalities

5. Page 4 (Line 88-90). I do not fully understand the analyses presented in these lines, maybe you can make that clearer.

6. Page 5 (Lines 94-98). A lot of statistical tests were used. Why e.g. did you use Fisher´s exact test? Are there situations where numbers are so small that a Chi-Square-Test should not be computed?

7. Page 5 (Lines 99-101). Did you use the raw number of deaths as the dependent variable or the number of deaths in relation to population size? If the raw number was used why was this the case? Because larger ZCTA´s should obviously experience more deaths.

7. Page 5 (Lines 101-104). This sentence is not correct and should be changed. A backward elimination is not limited to the presented family of models. The presented models are all models for count data. Why did you choose to investigate zero-inflated models that are appropriate for situations with an excess of zeros. Are there a lot of ZCTA´s with zero deaths? The distribution of the number of deaths per county would be interesting.

8. Page 6 (Line 119)- In Table 1 you have 2060 male mortalities in Table 2 2059.

9. Page 6 (Table 1). Why don´t you present percentages for categorical variables and standard deviations for continuous variables as outlined in the Methods section? Thereby one could see the differences between e.g. population by race and mortality by race. I would find an information on how many decedents have at least one (or/and two) comorbidities interesting, if the data contains this information. Is race the best wording or would ethnicity be better?

10. Page 7 (Lines 124-131, and Table 2). The text and the Table are hard to read, because you switch from row to column percentages. For example you write 777 of 1387 (56%) of white decedents are male, while in the Table a percentage of 37.7% is given for that number. The 56% is the number that makes the data easier to understand and allows a direct comparison between whites and minority decedents. The No category is not necessary for the comorbidities. Hypertensive Coronary Vascular Disease is also not statistically significant with the chosen significant level. I don´t understand the chosen statistical tests. There is only only one continuous outcome and two groups, thus a simple t-Test should work. Why did you do ANOVA and Kruskal-Wallis-Test and which p-value is reported? In my version there is a column parametric p-values and under the table the information that non-parametric p-values were computed, too. That´s confusing.

11. Page 8 (Line 141). In the brackets it should be stated what the effect measure is.

12. Page 8 (Line 142). How do you obtain these percentages throughout the paper? How does an IRR from 1.42 relate to the 58%?

13. Page 9 (Line 145). Here, you present four digits for the p-value, in the table two digits. In other tables three, it would be nicer to have the same value all the time. Marginally significant should not be used as a wording.

14. Page 9 (Table 3). A p-value of <.00 does not exist. A p-value of 0.07 should not be marked as significant when the significance level is chosen at 0.05. The IRR for Minority Zip as the independent variable and Number of Positive Results as the dependent variable (1.94) is greather than the IRR for the number of deaths (1.42). This should mean that the crude lethality rate is smaller for Minority dominated ZIP´s, although this might be explained by the lower average age at death in the minority group. Age would be an important confounder in these analyses, but I think it is not included in your county level dataset, right?

15. Page 9 (Line 152-153). It was not stated in the Methods section that an adjustment for other comorbidities is conducted.

16. Page 9/10 (Lines 154-163). A few of the Odds Ratios from Table 4 are presented in these lines. In the text 6 of the 8 reported OR are greater than 1, meaning that minorities have a greater chance to have one of the comorbidities than whites. In Table 4 the majority of OR is smaller than 1, therefore the text is not representing the Table adequately. Additionally, I don´t understand what these for other comorbidities adjusted Odds-Ratios should tell the reader. If some comorbidities are relatively more common among minorities others must be relatively less common, right? Therefore, an unadjusted Odds Ratio or a simple cross tabulation as in Table 2 would be more informative in my view. The labeling of the column “race” is not correct, the reference value should be stated secondly. E.g. it should be “Black Vs White” not “White Vs Black”. Some numbers differ between text and table. This should be checked.

17. Page 10 (Line 178). The reference to Figure 1 is misleading at this place, because median ages are not depicted in Figure 1.

18. Page 10 (Line 181). I would prefer not to present new results in the Discussion section, and don´t understand why you switch between median and average ages.

19. Page 10 (181-184). If this sentence is true, can be more easily assessed when the percentages in Table 2 are presented as suggested.

20. Page 11 (Lines 185-188). This explanation could have come earlier for me (Introduction). Until reading this sentence I didn´t know that this was your plan. Since your Introduction is rather short a more thorough outline of your plans would be helpful.

21. Page 13 (Line 217). The p-value presented in Table 3 is different.

Reviewer #2: Thank you for the opportunity to review Health Disparities and COVID-19: A Retrospective Study Examining Individual and Community Factors Causing Disproportionate COVID-19 Outcomes in Cook County, Illinois, March 16-May 31, 2020. This manuscript details an important examination of COVID-19 health disparities in Black and Brown communities, which is a topic that requires further attention in the literature. However, the manuscript in its current form has several limitations.

1. The introduction would benefit from a comprehensive review of the current COVID-19 health disparities literature and contextual grounding through a health disparities lens.

a. For example, the second paragraph states that “People of color, especially

those within Black and Latinx communities, are disproportionately affected by the disease.” The authors should provide a review of what is known, as the peer-reviewed literature regarding COVID-19 health disparities has grown substantially.

b. The introduction would also benefit from a contextual grounding of health disparities, perhaps within a social determinants of health framework, to orient the reader to the potential mechanisms that are exacerbating these glaring health disparities within Black and Brown communities. This would also help to ground the conclusions, wherein the authors enter into a discussion of structural inequities without a systematic framework for examining said inequities. A discussion of systemic racism would further strengthen this grounding. The following papers may provide useful reference:

Boyd RW, Lindo EG, Weeks LD, McLemore MR. On racism: a new standard for publishing on racial health inequities. Health Affairs Blog. Published online 2020.

Braveman P, Egerter S, Williams DR. The Social Determinants of Health: Coming of Age. Annual Review of Public Health. 2011;32(1):381-398. doi:10.1146/annurev-publhealth-031210-101218

Braveman P, Gottlieb L. The Social Determinants of Health: It’s Time to Consider the Causes of the Causes. Public Health Rep. 2014;129(1_suppl2):19-31. doi:10.1177/00333549141291S206

Link BG, Phelan J. Social Conditions As Fundamental Causes of Disease. Journal of Health and Social Behavior. 1995;35:80. doi:10.2307/2626958

Phelan JC, Link BG, Tehranifar P. Social Conditions as Fundamental Causes of Health Inequalities Theory, Evidence, and Policy Implications. Journal of Health and Social Behavior. 2010;51(1 suppl):S28-S40. doi:10.1177/0022146510383498

Poteat T, Millett G, Nelson LE, Beyrer C. Understanding COVID-19 Risks and Vulnerabilities among Black Communities in America: The Lethal Force of Syndemics. Ann Epidemiol. Published online May 14, 2020. doi:10.1016/j.annepidem.2020.05.004

c. The authors claim that “the current study described the demographic distribution of mortality and explores whether there is a causal relationship between race, neighborhood factors, and COVID-19 in Cook County, Illinois.” However, the manuscript does not appear to examine causal mechanisms of COVID-19 health disparities. Rather, there is an extensive examination of the associations between COVID-19 morbidity and mortality with race and ethnicity. This should be clarified upfront.

d. The authors also indicated that they examined “neighborhood factors,” but then provide ZIP Code and ZIP Code Tabulation Area data, which are not indicative of neighborhoods. Please make sure to use appropriate language to describe the geographic unit of analysis.

2. The methods section would benefit from additional clarification.

a. On lines 76-78, the authors provide a list of variables utilized in the ensuing analysis, but only use blanket terms (e.g., income, education). It would be helpful to specify what the variables are. For example, was median household income used in the analysis?

b. The authors indicated that “decedent’s zip codes were converted to ZIP Code Tabulation Areas,” on lines 79-80. Therefore, it would be beneficial to specify how this conversion was performed.

c. Following, on lines 81-82, the authors indicate that mortality data was aggregated by ZCTA and merged with ZCTA-level census data. How was this operation performed? Did the authors use geospatial software? This should be clarified.

d. On lines 83-84, the authors state that “minority predominant” ZCTAs were defined as having a minority population greater than the White population. For replication purposes, it would be helpful to specify percent minority used to calculate this variable. In addition, majority would be more accurate than “predominant,” particularly when referring to percent.

e. Please provide additional details regarding calculation of excess mortality (lines 85-87).

f. It would behoove the authors to clarify why they “compared trends between the top 20 ZCTAs” in the methods section (lines 88-90). This unfortunately doesn’t become clear until the discussion.

g. Lines 93-94—data is a plural noun, and should be conjugated appropriately.

h. Line 95—please use the correct test name—Kurskal-Wallis H

i. Line 97—"Chi-square tests was used…” also needs to be corrected for subject-verb agreement.

j. On lines 100-101, the authors indicate that they modeled mortality against risk factors. This is the first use of the language of risk, and it is not clear what these risks are. However, it is later (results) revealed that the authors considered race to be a risk factor. However, race (e.g., Black) is hardly a risk factor, whereas structural racism certainly is. Therefore, as stated above, the manuscript would benefit from a clear grounding of health disparities. There is a wealth of theory that provides this grounding—e.g., social determinants, fundamental causes, etc.

k. On lines 101-110, the authors provide explanations of various generalized linear modeling approaches, then only refer to their use of negative binomial regression in the ensuing text. Furthermore, no explanation of why NB was selected was offered. The manuscript would benefit from a concise explanation of why NB was selected (e.g., AIC values), while saving words to address the above comments (2. Methods a.-l.).

l. In addition, it would be beneficial to identify model composition in this section.

m. Please provide an ethics statement.

3. The results require further clarification.

a. On line 135, the authors state that “** The non-parametric p-value is calculated by the Kruskal-Wallis test for numerical covariates and Fisher's exact test for categorical covariates.” Yet, the “**” symbol does not appear to be provided in Table 2.

b. Lines 138-139 repeat the above stated issue, wherein it is not clear why the top 20 ZCTAs were examined.

c. Line 143—the language of risk factors requires amendment. Furthermore, it’s still unclear what these are.

d. Line 145—it would benefit the manuscript to describe what the authors consider a “marginally significant” range to be.

e. Line 142—please correct the following typo: “as a reference groups.”

f. Table 3 specifies composition of 6 models. It is curious that the authors did not develop multivariate models controlling for ZCTA-level variables—e.g., regressing number of deaths by ZCTA percent minority, poverty, household size, etc. At the very least, an explanation of why bivariate analysis was conducted would be useful. If appropriate, the analysis would benefit from extending these models to control for additional ZCTA-level factors.

g. The individual level findings are clear and concise.

h. Table 2 – one category is labeled “Diabetes Hypertension and Hypertension.” It’s unclear if the authors tested for both comorbidities or if this is a typo.

4. The discussion section would benefit from further clarification and additional examination of study findings.

a. Starting on line 189, the authors highlight potential associations between socioeconomic status and COVID-19 mortality, but only examined poverty in their models. Poverty alone is not an ideal indicator of SES, once again raising the question as to why other variables were not explored in multivariate models. For example, the authors collected income data (Median household income? This is still unclear.), yet did not indicate that they regressed morbidity or mortality by income or other SES-related variables. Education would be an important covariate in this context. The authors did mention in the methods section that ZCTA-level education and language data were collected, but these variables were not brought up again in the ensuing text.

b. On line 215, the authors contend that “household size is a statistically significant cause of COVID-19 mortality.” The provided analysis does not make a case for causality of household size. Furthermore, as mentioned above, the manuscript would benefit from a theoretical grounding that would help to unpack this association. It requires a careful analysis of why Black and Brown folk are at higher risk. A social determinants of health framework could be helpful in this context.

c. The language of cause should also be removed from the title.

d. One lines 221-222, the authors appropriately contend that viral transmission is not necessarily a function of population density, and recent studies have confirmed this. However, to reasonably understand this finding, the manuscript requires some theoretical grounding to orient readers to potential explanations of these findings. For example, while lower infection rates were observed in densely populated majority White ZCTAs, it is plausible that the difference here concerns access to health-protective resources associated with SES.

5. The conclusions would benefit from further refinement and clarification of terms.

a. The authors indicate that “the public health crisis” was “created by inbuilt barriers within minority communities.” It is not clear what these “inbuilt barriers” are. Are they referring to race, poverty? Race certainly is not a barrier, but unequal access to education and money resources, for example, certainly are. This argument provides further evidence of how a theoretical grounding would exponentially benefit this paper.

b. The authors are commended for arguing that COVID-19 health disparities are a function of structural inequities. This argument would be further strengthened by connecting this to structural racism, particularly given the findings of the current study.

c. On line 267, the authors reiterate that “race, more than SES, is the only statistically significant” factor associated with COVID-19 mortality. Again, as discussed above, developing integrated models may reveal additional factors. For example, income and education were never considered.

d. The argument concerning “structural inequality” on line 273 relates to points a and b above.

6. Please note that there are several typos throughout the manuscript.

6. PLOS authors have the option to publish the peer review history of their article (what does this mean?). If published, this will include your full peer review and any attached files.

Reviewer #1: No

Reviewer #2: No

---

## [Author Response · Author response to Decision Letter 0]

8 Feb 2021

RESPONSE: Using the provided style templates, the manuscript was formatted according to PLOS ONE’s formatting standards.

2. Please include the date(s) on which you accessed the databases or records to obtain the data used in your study.

RESPONSE: Dates that the databased were accessed for data collection were included in the materials and methods section.

3. We note that Table 1 appears to have a typographical error were one p-value is reported as "<.00" instead of "<.001". Please revise this.

 RESPONSE: The error typographical error in Table 3 (<.00) was changed to reflect the correct value (<.001)

4. PLOS ONE publication criteria and journal policy require authors to make all data underlying the findings described in their manuscript fully available without restriction, with rare exception (https://journals.plos.org/plosone/s/data-availability#loc-acceptable-data-access-restrictions). Please clarify whether all data used in your study is publicly available and how they can be accessed. We encourage authors to share de-identified or anonymized data if possible. For any third-party data that you cannot legally distribute, please include in the Data Availability Statement a description of the data set and all necessary third-party contact information others would need to apply to gain access to the data.

 RESPONSE: All data used in this manuscript are publicly available online. 

5. In your ethics statement in the Methods section and in the online submission form, please provide additional information about the data used in your retrospective study. Specifically, please ensure that you have discussed whether all data were fully anonymized before you accessed them. If not, please state whether your IRB waived the requirement for informed consent.

 RESPONSE: Statement of exemption was included at the end of the materials and methods section.

RESPONSE: Data can be accessed through: Soyemi, Kenneth (2021), “COVID 19 Mortality Disparity Data”, V1, doi: 10.17632/8dgz3pb73c.1

7. Please include a caption for figure 1.

RESPONSE: Caption was added to figure 1. 

Reviewer Comments and Author Responses:

Reviewer #1: Summary:

The authors investigate a very relevant question, the association between minority status and different Covid-19 related outcomes. The authors did a great work in integrating different data sources from the second largest county in the US. However, I see certain analyses that can be improved and do not agree in all their interpretations of the results. Specific comments and concerns are outlined below.

Comments:

1. Page 3 (Line 64). The authors say that they explore a causal relationship. I think a bit more cautious wording should be used with their study design.

Response: removed the verbiage implying causality

2. Page 4. I would find it helpful to make it even clearer (maybe subheadings) in which analyses the unit of investigation are persons and in which ZCTA´s.

Response: Subheadings describing individual data and ZCTA data were created.

3. Page 4 (Line 84). Why was the cut-off of 50% used for the ZCTA´s?

Response: Minority predominant ZCTA indicated ZCTAs where minorities (non-white individuals) outnumbered white individuals, and therefore were over 50% of the population.

4. Page 4 (Line 86). Moralities -> Mortalities

Response: Changed moralities to mortalities.

5. Page 4 (Line 88-90). I do not fully understand the analyses presented in these lines, maybe you can make that clearer.

Response: removed these sentences

6. Page 5 (Lines 94-98). A lot of statistical tests were used. Why e.g. did you use Fisher´s exact test? Are there situations where numbers are so small that a Chi-Square-Test should not be computed?

Response: removed the line about the Fischer exact test, because categorical covariates were not used

7. Page 5 (Lines 99-101). Did you use the raw number of deaths as the dependent variable or the number of deaths in relation to population size? If the raw number was used why was this the case? Because larger ZCTA´s should obviously experience more deaths.

Response: We used generalized linear modeling. The raw number of deaths was used but the number of deaths was adjusted in relation to population size through using offset of the count models.

7. Page 5 (Lines 101-104). This sentence is not correct and should be changed. A backward elimination is not limited to the presented family of models. The presented models are all models for count data. Why did you choose to investigate zero-inflated models that are appropriate for situations with an excess of zeros. Are there a lot of ZCTA´s with zero deaths? The distribution of the number of deaths per county would be interesting.

Response: We have changed and corrected the sentence. Regarding zero-inflated models: First, as see in Table 3, the outcome variables not only include the Number of Deaths, but also the Number of Residents Tested and the Number of Positive Results in the areas we studied. So the outcome variables could be zero-inflated. 

Second, we listed the candidates of count models including Poisson, negative binomial (NB), zero-inflated Poisson (ZIP), and zero-inflated negative binomial (ZINB) and compared them to choose final best fitted model(in this data, NB is the best model). The goal was to demonstrate that the final model we chose was based on appropriate modeling selection. 

8. Page 6 (Line 119)- In Table 1 you have 2060 male mortalities in Table 2 2059.

Response: Corrected Table 1 to 2059 male mortalities

9. Page 6 (Table 1). Why don´t you present percentages for categorical variables and standard deviations for continuous variables as outlined in the Methods section? Thereby one could see the differences between e.g. population by race and mortality by race. I would find an information on how many decedents have at least one (or/and two) comorbidities interesting, if the data contains this information. Is race the best wording or would ethnicity be better?

Response: Updated Tabe 1 to include the percentages and standard deviations. Added the rows outlining comorbidity numbers

10. Page 7 (Lines 124-131, and Table 2). The text and the Table are hard to read, because you switch from row to column percentages. For example you write 777 of 1387 (56%) of white decedents are male, while in the Table a percentage of 37.7% is given for that number. The 56% is the number that makes the data easier to understand and allows a direct comparison between whites and minority decedents. The No category is not necessary for the comorbidities. Hypertensive Coronary Vascular Disease is also not statistically significant with the chosen significant level. 

Response: Changes made per reviewer’s suggestion.

I don´t understand the chosen statistical tests. There is only one continuous outcome and two groups, thus a simple t-Test should work. Why did you do ANOVA and Kruskal-Wallis-Test and which p-value is reported? In my version there is a column parametric p-values and under the table the information that non-parametric p-values were computed, too. That´s confusing.

Response: we revised sentences in this reversion. Table 2 was extracted from the outputs of MACRO, which include continuous and categorical variables, and two and more than two group comparisons. Both parametric and non-parametric methods were used. All continuous variables are distributed normally and the number of cells are greater than 5. T-test and chi-square test are appropriate for continuous and categorical variables, respectively.

11. Page 8 (Line 141). In the brackets it should be stated what the effect measure is.

Response: Added IRR in the brackets

12. Page 8 (Line 142). How do you obtain these percentages throughout the paper? How does an IRR from 1.42 relate to the 58%?

Response: Removed the percentage

13. Page 9 (Line 145). Here, you present four digits for the p-value, in the table two digits. In other tables three, it would be nicer to have the same value all the time. Marginally significant should not be used as a wording.

Response: Tables 1,2 &3 (and throughout the paper) were updated so that p-values are consistently 4 decimals and CI are consistently 3 decimals

14. Page 9 (Table 3). A p-value of <.00 does not exist. A p-value of 0.07 should not be marked as significant when the significance level is chosen at 0.05. The IRR for Minority Zip as the independent variable and Number of Positive Results as the dependent variable (1.94) is greather than the IRR for the number of deaths (1.42). This should mean that the crude lethality rate is smaller for Minority dominated ZIP´s, although this might be explained by the lower average age at death in the minority group. Age would be an important confounder in these analyses, but I think it is not included in your county level dataset, right?

Response: Corrected <.00 to <0.001

Removed * from 0.07

It is correct that age was not included in the modelling, as our analysis included data aggregated to the zip code level. We modelled separately number of deaths vs residents from minority predominant zip codes, poverty, average household size at univariate. We did multivariate modelling deaths vs minority zip codes and poverty. IRR of 1.42 meant that residents from minority zip were 1.42 times likely to die when compared to residents from non-minority zip codes

15. Page 9 (Line 152-153). It was not stated in the Methods section that an adjustment for other comorbidities is conducted.

Removed the wording about adjustment for co-morbidities

16. Page 9/10 (Lines 154-163). A few of the Odds Ratios from Table 4 are presented in these lines. In the text 6 of the 8 reported OR are greater than 1, meaning that minorities have a greater chance to have one of the comorbidities than whites. In Table 4 the majority of OR is smaller than 1, therefore the text is not representing the Table adequately. Additionally, I don´t understand what these for other comorbidities adjusted Odds-Ratios should tell the reader. If some comorbidities are relatively more common among minorities others must be relatively less common, right? Therefore, an unadjusted Odds Ratio or a simple cross tabulation as in Table 2 would be more informative in my view. The labeling of the column “race” is not correct, the reference value should be stated secondly. E.g. it should be “Black Vs White” not “White Vs Black”. Some numbers differ between text and table. This should be checked.

Response: Appreciate the reviewer’s comments.

Table was changed to reflect Black vs White. 

Black vs White and Latin X vs White and other Race vs White corrected throughout the table. Table 4 now reads as using white decendents as a reference group. Our multivariate analysis revealed several notable results. Of decendents with diabetes, Black individuals were 64% more likely to die of COVID-19 [Odds Ratio OR=1.37 (1.16, 1.62)], Latinx decendents were over two times as likely to die [OR=2.11(1.72, 2.60)], and other races were 63% more likely to die compared with white individuals. Of obese decendents, Black individuals were 49% more likely to die from COVID-19 [OR=1.52(1.04, 2.21)], Latinx individuals were again over to two times as likely to die [OR=2.17(1.43,3.29)], but oter races were 80% less likely to die compared with obese White individuals [OR=0.20(0.03, 1.43)]. OF decedents with hypertension, Black individuals were 62% more likely to die of COVID-19 [OR=1.38 (1.15, 1.66)], other combined races were 70% more likely to die [OR=1.29 (0.87, 1.93)], but Latinx individuals were 37% less likely to die than White individuals [OR=0.63(0.51,0.77)]. (table 4)

17. Page 10 (Line 178). The reference to Figure 1 is misleading at this place, because median ages are not depicted in Figure 1.

Response: moved the reference to the figure.

18. Page 10 (Line 181). I would prefer not to present new results in the Discussion section, and don´t understand why you switch between median and average ages.

Response: removed this paragraph as it is new data

19. Page 10 (181-184). If this sentence is true, can be more easily assessed when the percentages in Table 2 are presented as suggested.

Response: Completed percentages added to Table 2

20. Page 11 (Lines 185-188). This explanation could have come earlier for me (Introduction). Until reading this sentence I didn´t know that this was your plan. Since your Introduction is rather short a more thorough outline of your plans would be helpful.

Response: Added mention of the 5 possible contributing factors we explored under the introduction 

21. Page 13 (Line 217). The p-value presented in Table 3 is different.

Response: Corrected to 0.049

Reviewer #2: Thank you for the opportunity to review Health Disparities and COVID-19: A Retrospective Study Examining Individual and Community Factors Causing Disproportionate COVID-19 Outcomes in Cook County, Illinois, March 16-May 31, 2020. This manuscript details an important examination of COVID-19 health disparities in Black and Brown communities, which is a topic that requires further attention in the literature. However, the manuscript in its current form has several limitations.

1. The introduction would benefit from a comprehensive review of the current COVID-19 health disparities literature and contextual grounding through a health disparities lens.

a. For example, the second paragraph states that “People of color, especially

those within Black and Latinx communities, are disproportionately affected by the disease.” The authors should provide a review of what is known, as the peer-reviewed literature regarding COVID-19 health disparities has grown substantially.

b. The introduction would also benefit from a contextual grounding of health disparities, perhaps within a social determinants of health framework, to orient the reader to the potential mechanisms that are exacerbating these glaring health disparities within Black and Brown communities. This would also help to ground the conclusions, wherein the authors enter into a discussion of structural inequities without a systematic framework for examining said inequities. A discussion of systemic racism would further strengthen this grounding. 

Response: Introduction updated to reflect the current literature on health disparities and people of color.

c. The authors claim that “the current study described the demographic distribution of mortality and explores whether there is a causal relationship between race, neighborhood factors, and COVID-19 in Cook County, Illinois.” However, the manuscript does not appear to examine causal mechanisms of COVID-19 health disparities. Rather, there is an extensive examination of the associations between COVID-19 morbidity and mortality with race and ethnicity. This should be clarified upfront.

Response: Removed the wording suggesting causality 

d. The authors also indicated that they examined “neighborhood factors,” but then provide ZIP Code and ZIP Code Tabulation Area data, which are not indicative of neighborhoods. Please make sure to use appropriate language to describe the geographic unit of analysis.

Response: Changed mention of neighborhood factors to zip code factors

2. The methods section would benefit from additional clarification.

a. On lines 76-78, the authors provide a list of variables utilized in the ensuing analysis, but only use blanket terms (e.g., income, education). It would be helpful to specify what the variables are. For example, was median household income used in the analysis?

Response: Changed income to average ZCTA population income

Changed education to education data including percent who obtained high school and bachelor’s degrees

Changed poverty to percent of ZCTA above poverty line

Changed language to primary language.

b. The authors indicated that “decedent’s zip codes were converted to ZIP Code Tabulation Areas,” on lines 79-80. Therefore, it would be beneficial to specify how this conversion was performed.

Response: Changed wording to “decedents zip codes were substituted for ZCTA” and added “and in Chicago represent the same spatial areas”

c. Following, on lines 81-82, the authors indicate that mortality data was aggregated by ZCTA and merged with ZCTA-level census data. How was this operation performed? Did the authors use geospatial software? This should be clarified.

Response: Clarified that Microsoft Excel was used to merge the mortality data with the census data by ZCTA

d. On lines 83-84, the authors state that “minority predominant” ZCTAs were defined as having a minority population greater than the White population. For replication purposes, it would be helpful to specify percent minority used to calculate this variable. In addition, majority would be more accurate than “predominant,” particularly when referring to percent.

Response: Changed the wording to reflect that minority predominant ZCTA was defined as one where the non-white population was over 50% of the population.

e. Please provide additional details regarding calculation of excess mortality (lines 85-87).

Response: We removed the statement containing excess mortality

f. It would behoove the authors to clarify why they “compared trends between the top 20 ZCTAs” in the methods section (lines 88-90). This unfortunately doesn’t become clear until the discussion.

Response: Removed the mention of the comparison of the top 20 ZCTAs.

g. Lines 93-94—data is a plural noun, and should be conjugated appropriately.

Response: Changed Data to Datum

h. Line 95—please use the correct test name—Kurskal-Wallis H

Response: corrected test name 

i. Line 97—"Chi-square tests was used…” also needs to be corrected for subject-verb agreement.

Response: Changed to Chi-Square tests were used.

j. On lines 100-101, the authors indicate that they modeled mortality against risk factors. This is the first use of the language of risk, and it is not clear what these risks are. However, it is later (results) revealed that the authors considered race to be a risk factor. However, race (e.g., Black) is hardly a risk factor, whereas structural racism certainly is. Therefore, as stated above, the manuscript would benefit from a clear grounding of health disparities. There is a wealth of theory that provides this grounding—e.g., social determinants, fundamental causes, etc.

Response: Removed risk factor wording.

k. On lines 101-110, the authors provide explanations of various generalized linear modeling approaches, then only refer to their use of negative binomial regression in the ensuing text. Furthermore, no explanation of why NB was selected was offered. The manuscript would benefit from a concise explanation of why NB was selected (e.g., AIC values), while saving words to address the above comments (2. Methods a.-l.).

Response: Changed this section to better reflect our analysis.

l. In addition, it would be beneficial to identify model composition in this section.

Response: added in the model composition

m. Please provide an ethics statement.

Response: Ethics statement included

3. The results require further clarification.

a. On line 135, the authors state that “** The non-parametric p-value is calculated by the Kruskal-Wallis test for numerical covariates and Fisher's exact test for categorical covariates.” Yet, the “**” symbol does not appear to be provided in Table 2.

Response: Removed the footer with the **symbol as it was not used.

b. Lines 138-139 repeat the above stated issue, wherein it is not clear why the top 20 ZCTAs were examined.

Response: Removed references to the top 20 ZCTAs

c. Line 143—the language of risk factors requires amendment. Furthermore, it’s still unclear what these are.

Response: Removed the word risk, now it is factors

d. Line 145—it would benefit the manuscript to describe what the authors consider a “marginally significant” range to be.

Response: Removed wording of marginal significance

e. Line 142—please correct the following typo: “as a reference groups.”

Response: Now reads “as a reference group”

f. Table 3 specifies composition of 6 models. It is curious that the authors did not develop multivariate models controlling for ZCTA-level variables—e.g., regressing number of deaths by ZCTA percent minority, poverty, household size, etc. At the very least, an explanation of why bivariate analysis was conducted would be useful. If appropriate, the analysis would benefit from extending these models to control for additional ZCTA-level factors.

Response:The initial model was controlled for ZCTA percent minority, race, poverty and house size. We added the information in this reversion. The results were reported based on final model after removing the non-significant terms.

g. The individual level findings are clear and concise.

Response: Thank you

h. Table 2 – one category is labeled “Diabetes Hypertension and Hypertension.” It’s unclear if the authors tested for both comorbidities or if this is a typo.

Response: Clarified that this variable was people who had triple disease ( diabetes, hypertension, and obesity)

4. The discussion section would benefit from further clarification and additional examination of study findings.

a. Starting on line 189, the authors highlight potential associations between socioeconomic status and COVID-19 mortality, but only examined poverty in their models. Poverty alone is not an ideal indicator of SES, once again raising the question as to why other variables were not explored in multivariate models. For example, the authors collected income data (Median household income? This is still unclear.), yet did not indicate that they regressed morbidity or mortality by income or other SES-related variables. Education would be an important covariate in this context. The authors did mention in the methods section that ZCTA-level education and language data were collected, but these variables were not brought up again in the ensuing text.

Response: We agree that Poverty alone is not indicative of SES. But poverty is a known risk factor for SES. We did not have individual SES for decedants. We coded for either above or below the FDL. Based on this coding we used this for SES.The text was corrected according to the modeling that we used

b. On line 215, the authors contend that “household size is a statistically significant cause of COVID-19 mortality.” The provided analysis does not make a case for causality of household size. Furthermore, as mentioned above, the manuscript would benefit from a theoretical grounding that would help to unpack this association. It requires a careful analysis of why Black and Brown folk are at higher risk. A social determinants of health framework could be helpful in this context.

Response: Changed the causal wording to reflect that it is a contributing factor

c. The language of cause should also be removed from the title.

Response: Removed Causal from the title

d. One lines 221-222, the authors appropriately contend that viral transmission is not necessarily a function of population density, and recent studies have confirmed this. However, to reasonably understand this finding, the manuscript requires some theoretical grounding to orient readers to potential explanations of these findings. For example, while lower infection rates were observed in densely populated majority White ZCTAs, it is plausible that the difference here concerns access to health-protective resources associated with SES.

Response: This was addressed in the introduction.

5. The conclusions would benefit from further refinement and clarification of terms.

a. The authors indicate that “the public health crisis” was “created by inbuilt barriers within minority communities.” It is not clear what these “inbuilt barriers” are. Are they referring to race, poverty? Race certainly is not a barrier, but unequal access to education and money resources, for example, certainly are. This argument provides further evidence of how a theoretical grounding would exponentially benefit this paper.

Response: Clarified inbuilt barriers 

b. The authors are commended for arguing that COVID-19 health disparities are a function of structural inequities. This argument would be further strengthened by connecting this to structural racism, particularly given the findings of the current study.

Response: Added information to the introduction to support our discussion on structural racism.

c. On line 267, the authors reiterate that “race, more than SES, is the only statistically significant” factor associated with COVID-19 mortality. Again, as discussed above, developing integrated models may reveal additional factors. For example, income and education were never considered.

Response: We had considered multiple variables including income and education as coviariates in our multivariate regression modeling. We removed the non-significant variables in the final model using backward model selection.

d. The argument concerning “structural inequality” on line 273 relates to points a and b above.

Response: Addressed as per our response to points a and b.

6. Please note that there are several typos throughout the manuscript.

Response: Corrected all identified typos.

---

## [Decision Letter · Decision Letter 1]

17 Mar 2021

PONE-D-20-25294R1

Health Disparities and COVID-19: A Retrospective Study Examining Individual and Community Factors Causing Disproportionate COVID-19 Outcomes in Cook County, Illinois, March 16-May 31, 2020

PLOS ONE

Dear Dr. Soyemi,

Thank you for submitting your revised manuscript to PLOS ONE. After careful consideration, we feel that it has merit but still does not fully meet PLOS ONE’s publication criteria as it currently stands. We invite you to submit another revised version of the manuscript that addresses the points raised during the review process.

There remain numerous inconsistencies and inaccurarcies with respect to the numerical data presented. This clearly impacts on the scientific quality and must be adressed. Please ensure that all numbers and measures fully correspond between tables and text (see reviewer 1 comments), and digits are used in a consistent manner. Furthermore, the argumentation refering to social deterninants of health should be strengthened with some more explanation and references, a core point of reviewer 2 comments.

We look forward to receiving your revised manuscript.

Kind regards,

Hajo Zeeb

Academic Editor

PLOS ONE

Reviewers' comments:

Reviewer's Responses to Questions

**Comments to the Author**

1. If the authors have adequately addressed your comments raised in a previous round of review and you feel that this manuscript is now acceptable for publication, you may indicate that here to bypass the “Comments to the Author” section, enter your conflict of interest statement in the “Confidential to Editor” section, and submit your "Accept" recommendation.

Reviewer #1: (No Response)

Reviewer #2: (No Response)

2. Is the manuscript technically sound, and do the data support the conclusions?

Reviewer #1: No

Reviewer #2: Yes

3. Has the statistical analysis been performed appropriately and rigorously? 

Reviewer #1: No

Reviewer #2: Yes

4. Have the authors made all data underlying the findings in their manuscript fully available?

Reviewer #1: Yes

Reviewer #2: Yes

5. Is the manuscript presented in an intelligible fashion and written in standard English?

Reviewer #1: Yes

Reviewer #2: Yes

6. Review Comments to the Author

Reviewer #1: Summary: The authors improved the manuscript over the previous version. However, I need to say that there are still a lot of issues with the manuscript. As I very much agree with their conclusion in general, I would encourage the authors to improve their writing and pay more attention to details. There are still too many inconsictencies in the paper for me to accept it in the current state.

Comments:

Abstract:

1a) (Lines 57-59): I would prefer that you write out IRR in the abstract, because it´s not a very often used abbreviation. p-Values differ in their number of digits, that should be changed.

Introduction:

2a) (Lines 81-83): The wording is to strong for me. There is a rich body of scientific literature that links race/ethnicity to health outcomes. Maybe it´s more the awareness in society and the transmission of the scientific knowledge that is lacking. And in the next sentence you cite five papers linking social determinants and health outcomes.

2b) (Lines 96-100): Are these numbers confidence intervals and point estimates mixed? This part is confusing.

Materials and methods:

3a) (Line 116): Individual analysis -> Individual level analysis

3b) (Lines 132-135): Percent below the federal povery line -> proportion of persons living below the federal povery line. The same for line 134.

3c) (Lines 165-172): You used a lot of model fit indices. Although I would not proceed as you did, I can agree on that. But the chosen model should not be part of the methods section. Move to results section.

3d) (Lines 173-176): You write that you compare comorbidities of different ethnic groups. It´s the lethality of Covid-19 in different comorbidity groups, right? Make the outcome clearer.

3d) (Lines 177-178): Can you elaborate why this sentence stands there out of context after the statistical analysis?

Results and discussion;

4a) (Lines 183-185): In Table 1 this number is 59%. Be consistent with your numbers. Mean is 73.8 and SD 14.73. The same number of digits should be used. The same applies for Line 185 and other parts of the text.

4b) (Table 1): The unkown category for cook county population by sex can be deleted as there are no unknown persons.

4c) (Lines 193-195): You still write 777 of 1387 (56%), but in Table 2 you present a number of 37.7%. Giving two different percentages for the same number is very confusing. Table 2 would be so much easier to read, if you would present the 56% and do it accordingly for all values in the Table.

4d) (Line 210): Give an abbreviation for the effect measure you present here (IRR). Additionally, the numbers in the text and Table 3 are not the same.

4e) (Lines 219-229): It is confusing to write 64% more likely and present an Odds Ratio of 1.37. How do you obtain the number 64% more likely? The same applies for the next sentences. Also, the interpretation of these results is not reflecting the Table very well, because you mainly interpret the Odds Ratios larger than 1. This whole part is very hard to read and confusing. I would recommend to exclude these lines and simply add columns to Table 2 and present the correct percentages for the comorbidities there.

4f) (Lines 271-273): “56% or 1.4 more likely to die from Covid-19 …..(IRR=1.44)”. You confuse the reader by all this different numbers.

4g) (Lines 277-279): significantly higher chances (higher is missing). You reference to Table 3, that is not correct.

4h) (Line 344): Delete to.

Reviewer #2: Thank you for the opportunity to review the revised version of Health Disparities and COVID-19: A Retrospective Study Examining Individual and Community Factors Causing Disproportionate COVID-19 Outcomes in Cook County, Illinois, March 16-May 31, 2020. The authors have generally addressed my comments, and the manuscript has improved. I am providing some additional feedback before submitting a publication recommendation.

• I appreciate the inclusion of social determinants of health. However, the introduction requires additional context regarding how the social determinants of health may contribute to health disparities.

• The third paragraph cites one study indicating that “differences in healthcare access and exposure risk” may contribute to Black and Latinx COVID-19 disparities, but there’s no explanation of why Black and Latinx persons experience these resource inequities. The social determinants of health, on which there’s a wealth of literature (see the cited Braveman manuscript), details these challenges. The current manuscript would benefit from a more careful application of the social determinants of health.

• Line 85 – it’s unclear what ZCTA-level associations the authors are interested in. Poverty, household size?

• Line 96 – please include the specific decedent demographics.

• Variables used as indicators for the social determinants need to be explicitly detailed in the methods.

• Lines 108 – 110 – please explain which variables are social determinants.

• A rational for not including income in the models would be helpful. That is, average household size, poverty, income, etc. were obtained from 2018 ACS, yet income was not explored.

• The discussion section should be clearly labeled.

• Associations between several variables and COVID outcomes were explained in the discussion without making explicit connection to social determinants of health. I would like to recommend amending this section to reconnect with social determinants.

• As discussed, with poverty as the only indicator of SES, it should be accordingly be labeled poverty, not SES. For example, the authors argue that race is a stronger indicator of COVID-19 mortality than SES. Yet, according to their analyses, race is a stronger indicator than poverty.

• Finally, there are still several typos.

7. PLOS authors have the option to publish the peer review history of their article (what does this mean?). If published, this will include your full peer review and any attached files.

Reviewer #1: No

Reviewer #2: No

---

## [Author Response · Author response to Decision Letter 1]

12 Oct 2021

We have added new data and addressed all issues raised by the reviewers

---

## [Decision Letter · Decision Letter 2]

4 Feb 2022

PONE-D-20-25294R2Health Disparities and COVID-19: A Retrospective Study Examining Individual and Community Factors Causing Disproportionate COVID-19 Outcomes in Cook County, Illinois.PLOS ONE

Dear Dr. Soyemi,

Thank you for submitting your manuscript to PLOS ONE. After careful consideration, we feel that it has merit but does not fully meet PLOS ONE’s publication criteria as it currently stands. Therefore, we invite you to submit a revised version of the manuscript that addresses the points raised during the review process.

We look forward to receiving your revised manuscript.

Kind regards,

Gaetano Santulli, MD

Academic Editor

PLOS ONE

Reviewers' comments:

Reviewer's Responses to Questions

**Comments to the Author**

1. If the authors have adequately addressed your comments raised in a previous round of review and you feel that this manuscript is now acceptable for publication, you may indicate that here to bypass the “Comments to the Author” section, enter your conflict of interest statement in the “Confidential to Editor” section, and submit your "Accept" recommendation.

Reviewer #1: (No Response)

Reviewer #3: All comments have been addressed

2. Is the manuscript technically sound, and do the data support the conclusions?

Reviewer #1: Partly

Reviewer #3: Partly

3. Has the statistical analysis been performed appropriately and rigorously? 

Reviewer #1: No

Reviewer #3: Yes

4. Have the authors made all data underlying the findings in their manuscript fully available?

Reviewer #1: No

Reviewer #3: Yes

5. Is the manuscript presented in an intelligible fashion and written in standard English?

Reviewer #1: No

Reviewer #3: Yes

6. Review Comments to the Author

Reviewer #1: General comments:

The authors replied to the reviewer comments and rewrote large parts of the manuscript.

Firstly, I want to mention the parts that I really liked. The data were updated and now span nearly 1,5 years of the pandemic. Thereby, greater precision could be achieved. The visualization of a newly calculated interaction effect made the results intuitively visible, and the conclusion is beautifully written and I very much agree with their view the way society must be improved to create better health outcomes.

However, the authors introduced a lot of new mistakes. Introductory statements about associations without reference, wrongly calculated numbers, the introduction of an interaction effect into the analysis and risk differences/relative risks without mentioning it in the methods section, and a lot of typos are the most obvious.

Therefore, I recommend to the editor to give the authors a last chance to get their manuscript in a form that is publishable. The consultation of a biostatistician/epidemiologist could be essential to get their numbers right. I need to say that I find it very uncommon that a third round of review contains so many mistakes. In future work, I wish the authors would employ a greater scrutiny to their work.

Specific comments:

Introduction:

1a) “There is a causal link between social factors, health indicators, and measures of individuals’ socioeconomic resources, such as income, educational attainment, and social position. This association simulates a stepwise gradient pattern, with health improving incrementally as social position rises. This relationship is dosedependent and is predominant in Non-Hispanic Black and Non-Hispanic White groups compared with Latinx populations.”

Comment: A reference for these sentences is necessary.

1b) “However, as is true with many diseases, COVID-19 outcomes are strongly associated with race.”

Comment: A reference is necessary.

1c) “A systematic review by Mackay et al.”

Comment: After such a sentence the reference should appear. Additionally, this seems to be reference 9 (Mackey et al.).

1d) “Similarly, same study showed that Latin X…”

Comment: English should be used properly.

Methods:

2a) per capital

Comment: per capita

Results:

3a) “Mean and standard deviation age at death was lower for Non-Whites compared with

Whites 70 (15) vs 78 (13); p < 0.01.“

Comment: Only the mean was lower.

3b) “risk difference of 0.03 [(95% CI: 0.15, 0.53); P = <0.01)]”

Comment: Firstly, a point estimate has to be part of the confidence interval. Secondly, you don´t write in the methods section that you calculated risk differences.

3c) “0.11 [(95% CI: -0.07, 0.30, 0.17); P = <0.20)] for hypertension“

Comment: There are three numbers in the confidence interval. Additionally, you report 60% of the non-white and 59% of the white decedents having a hypertension in Table 2. Thus, the risk difference is 0.01. There are more of these mistakes in Table 2.

3d) “0.04 [(95% CI: 0.03, 0.58); P = <0.01) for obesity.”

Comment: This confidence interval seems to be incorrect. Confidence intervals for risk differences are usually symmetric around the point estimate.

3e) “Direct comparison of COVID-19 associated morbidities for Non-White compared with White decedents with hypertension showed little difference [Relative Risk (RR) 1.01, 95% CI

(0.98, 1.05)].”

Comment: If the confidence interval includes 1 they showed no difference.

3f) “In the final model after adjusting for factors described above, residents of Non-White ZCTAs were 1.77 times as likely to die from COVID-19 [IRR 1.77, 95% CI (1.17, 2.66)]. Residents in households with a high percentage of Spanish speakers had increased risk of testing positive [IRR 1.69, 95% CI (1.03, 2.75)]. Living in poverty alone was not associated with an increased number of deaths, but the interaction of living in poverty and in a Non-White ZCTA increased the risk 2 to 3 times [IRR 2.99, 95% CI (0.71, 12.57)].”

Comment: In the previous version you didn´t include interactions. Although, the interaction is not significant, it seems relevant. But at least, tell the reader in the statistical analysis section that you investigated this interaction.

Discussion:

4a) Comment: I don´t know about the exact guidelines of Plos One, but I find it uncommon to present new results (Figure 2) in the discussion section.

4b) Comment: I think you cannot say COVID-19 infection. It should be COVID-19 or SARS-CoV-2-infection.

4c) “As we compared comorbidities between ethnic/racial groups, we found that minority COVID-19 decedents had statistically significant chances of having comorbidities at the time of death.”

Comment: Significantly higher?

Reviewer #3: WELL WRITTEN ARTICLE,

I ALSO VERY APPRECIATED THE CONCLUSION AND REFLECTION OF THE STRUCTURAL INEQUALITY WITH A CONSEQUENT DECREASE OF OPPORTUNITIES.

The following pertinent reports should be mentioned/discussed:

PMID: 32374952

PMID: 32293639

PMID: 32305087

PMID: 35112280

PMID: 35020749

PMID: 34997001

PMID: 34836206

PMID: 34762110

PMID: 33519709

7. PLOS authors have the option to publish the peer review history of their article (what does this mean?). If published, this will include your full peer review and any attached files.

Reviewer #1: No

Reviewer #3: **Yes: **CAROLINA BOLOGNA

---

## [Author Response · Author response to Decision Letter 2]

10 Apr 2022

Dear Editors and Reviewers,

Thank you for your time and consideration in reviewing our manuscript. We appreciate your

feedback. All comments have been addressed. Please see below for our responses.

Comments:

Introduction 

Abstract:

1a) “There is a causal link between social factors, health indicators, and measures of individuals’ socioeconomic resources, such as income, educational attainment, and social position. This association simulates a stepwise gradient pattern, with health improving incrementally as social position rises. This relationship is dose dependent and is predominant in Non-Hispanic Black and Non-Hispanic White groups compared with Latinx populations.”

Comment: A reference for these sentences is necessary.

Response: added citation [9] after that sentence

1b) “However, as is true with many diseases, COVID-19 outcomes are strongly associated with race.”

Comment: A reference is necessary.

Response: added citations 13-17 

1c) “A systematic review by Mackay et al.”

Comment: After such a sentence the reference should appear. Additionally, this seems to be reference 9 (Mackey et al.).

Response: added citation [18] after that sentence

1d) “Similarly, same study showed that Latin X…”

Comment: English should be used properly.

Response: added the word “the” 

Now reads :Similarly, the same study showed that Latin X

Methods:

2a) per capital

Comment: per capita

Response: removed the l from capital to make it per capita

Results:

3a) “Mean and standard deviation age at death was lower for Non-Whites compared with

Whites 70 (15) vs 78 (13); p < 0.01.“

Comment: Only the mean was lower.

Response: removed “ and standard deviation” from this sentence

3b) “risk difference of 0.03 [(95% CI: 0.15, 0.53); P = <0.01)]”

Comment: Firstly, a point estimate has to be part of the confidence interval. Secondly, you don´t write in the methods section that you calculated risk differences.

“risk difference of 0.03 [(95% CI: 0.15, 0.53); P = <0.01)]” Firstly, a point estimate has to be part of the confidence interval.

Response: Corrected 0.03 (0.02,0.05)

Secondly, you don´t write in the methods section that you calculated risk differences.

Response: It was in the methods and was written as “We also reported the Relative Risk (RR) and risk differences and their corresponding 95% confidence intervals”.

3c) “0.11 [(95% CI: -0.07, 0.30, 0.17); P = <0.20)] for hypertension“

Comment: There are three numbers in the confidence interval. 

Response: corrected to 0.11 [(95% CI: -0.07, 0.30); P = <0.20)] for hypertension 

Additionally, you report 60% of the non-white and 59% of the white decedents having a hypertension in Table 2. Thus, the risk difference is 0.01. There are more of these mistakes in Table 2.

Corrected now reads 60% of the non-white and 59% of the white decedents had hypertension in Table 2. Thus, the risk difference is 0.01.

3d) “0.04 [(95% CI: 0.03, 0.58); P = <0.01) for obesity.”

Comment: This confidence interval seems to be incorrect. Confidence intervals for risk differences are usually symmetric around the point estimate.

Response: Corrected now reads 0.03 (0.02 ,0.05)

3e) “Direct comparison of COVID-19 associated morbidities for Non-White compared with White decedents with hypertension showed little difference [Relative Risk (RR) 1.01, 95% CI

(0.98, 1.05)].”

Comment: If the confidence interval includes 1 they showed no difference.

Response: changed “little” to “no”

3f) “In the final model after adjusting for factors described above, residents of Non-White ZCTAs were 1.77 times as likely to die from COVID-19 [IRR 1.77, 95% CI (1.17, 2.66)]. Residents in households with a high percentage of Spanish speakers had increased risk of testing positive [IRR 1.69, 95% CI (1.03, 2.75)]. Living in poverty alone was not associated with an increased number of deaths, but the interaction of living in poverty and in a Non-White ZCTA increased the risk 2 to 3 times [IRR 2.99, 95% CI (0.71, 12.57)].”

Comment: In the previous version you didn´t include interactions. Although, the interaction is not significant, it seems relevant. But at least, tell the reader in the statistical analysis section that you investigated this interaction.

Response: We added to the methods the following: For ZCTA analysis, we created one model for each outcome variable (mortality, number of positive tests and total number tested per ZCTA). For the mortality (deaths) per ZCTA, we used four models with sequential addition of economic and housing population covariates not included in the SDI. after each round of modelling To understand the relationships between the variables we added the interaction term of Non-White (Minority Zip) ZCTA and if greater than 20% of households that lived in poverty.

Discussion:

4a) Comment: I don´t know about the exact guidelines of Plos One, but I find it uncommon to present new results (Figure 2) in the discussion section.

Response: – moved the table out of discussion section

4b) Comment: I think you cannot say COVID-19 infection. It should be COVID-19 or SARS-CoV-2-infection.

Response: removed the word infection

4c) “As we compared comorbidities between ethnic/racial groups, we found that minority COVID-19 decedents had statistically significant chances of having comorbidities at the time of death.”

Comment: Significantly higher?

Response: changed to “ higher likelihood of

Reviewer #3: WELL WRITTEN ARTICLE,

I ALSO VERY APPRECIATED THE CONCLUSION AND REFLECTION OF THE STRUCTURAL INEQUALITY WITH A CONSEQUENT DECREASE OF OPPORTUNITIES.

The following pertinent reports should be mentioned/discussed:

PMID: 32374952 Racial Health Disparities and Covid-19 - Caution and Context N Engl J Med 2020 Jul 16;383(3):201-203. doi: 10.1056/NEJMp2012910. Epub 2020 May 6.

PMID: 32293639 COVID-19 and African Americans JAMA 2020 May 19;323(19):1891-1892. doi: 10.1001/jama.2020.6548

PMID: 32305087 COVID-19 exacerbating inequalities in the US Lancet 2020 Apr 18;395(10232):1243-1244. doi: 10.1016/S0140-6736(20)30893-X.

PMID: 35112280 Utilization Gaps During the COVID-19 Pandemic: Racial and Ethnic Disparities in Telemedicine Uptake in Federally Qualified Health Center Clinic J Gen Intern Med 2022 Feb 2;1-7. doi: 10.1007/s11606-021-07304-4. Online ahead of print.

PMID: 35020749 Increased mask adherence after important politician infected with COVID-19 PLoS One 2022 Jan 12;17(1):e0261398. doi: 10.1371/journal.pone.0261398. eCollection 2022.

PMID: 34997001 Geographic and temporal variation in racial and ethnic disparities in SARS-CoV-2 positivity between February 2020 and August 2021 in the United States Sci Rep 2022 Jan 7;12(1):273. doi: 10.1038/s41598-021-03967-5.

PMID: 34836206 l-Arginine and COVID-19: An Update Nutrients 2021 Nov 5;13(11):3951. doi: 10.3390/nu13113951.

PMID: 34762110 Disparities in COVID-19 Outcomes by Race, Ethnicity, and Socioeconomic Status: A Systematic-Review and Meta-analysis JAMA Netw Open. 2021 Nov 1;4(11):e2134147. doi: 10.1001/jamanetworkopen.2021.34147.

PMID: 33519709 Metformin Use Is Associated With Reduced Mortality in a Diverse Population With COVID-19 and DiabetesFront Endocrinol (Lausanne) . 2021 Jan 13;11:600439. doi: 10.3389/fendo.2020.600439. eCollection 2020

Response: We replaced some of the previous references with the suggested references

---

## [Decision Letter · Decision Letter 3]

28 Apr 2022

Health Disparities and COVID-19: A Retrospective Study Examining Individual and Community Factors Causing Disproportionate COVID-19 Outcomes in Cook County, Illinois.

PONE-D-20-25294R3

Dear Dr. Soyemi,

We’re pleased to inform you that your manuscript has been judged scientifically suitable for publication and will be formally accepted for publication once it meets all outstanding technical requirements.

Kind regards,

Gaetano Santulli, MD

Academic Editor

PLOS ONE

Additional Editor Comments (optional):

Reviewers' comments:

Reviewer's Responses to Questions

**Comments to the Author**

1. If the authors have adequately addressed your comments raised in a previous round of review and you feel that this manuscript is now acceptable for publication, you may indicate that here to bypass the “Comments to the Author” section, enter your conflict of interest statement in the “Confidential to Editor” section, and submit your "Accept" recommendation.

Reviewer #1: All comments have been addressed

2. Is the manuscript technically sound, and do the data support the conclusions?

Reviewer #1: Partly

3. Has the statistical analysis been performed appropriately and rigorously? 

Reviewer #1: Yes

4. Have the authors made all data underlying the findings in their manuscript fully available?

Reviewer #1: Yes

5. Is the manuscript presented in an intelligible fashion and written in standard English?

Reviewer #1: Yes

6. Review Comments to the Author

Reviewer #1: The authors addressed the comments from the previous revision. I think the paper is now in a form that is publishable.

7. PLOS authors have the option to publish the peer review history of their article (what does this mean?). If published, this will include your full peer review and any attached files.

Reviewer #1: No

---

## [Editor Report · Acceptance letter]

6 May 2022

PONE-D-20-25294R3 

Health disparities and COVID-19: A retrospective study examining individual and community factors causing disproportionate COVID-19 outcomes in Cook County, Illinois 

Dear Dr. Soyemi:

I'm pleased to inform you that your manuscript has been deemed suitable for publication in PLOS ONE. Congratulations! Your manuscript is now with our production department. 

Kind regards, 

on behalf of

Professor Gaetano Santulli 

Academic Editor

PLOS ONE